# Prevalence, predictors, and economic burden of mental health disorders among asylum seekers, refugees and migrants from African countries: A scoping review

**Wael Osman**[1,2,3]*, **France Ncube**[3,4], **Kamil Shaaban**[5,6,7], **Alaa Dafallah**[5,8]

1 Department of Biological Sciences, College of Medicine and Health Sciences, Khalifa University, Abu Dhabi, United Arab Emirates, 2 Centre of Biotechnology, Khalifa University, Abu Dhabi, United Arab Emirates, 3 Liverpool John Moores University in Partnership with UNICAF University, Liverpool, United Kingdom, 4 Lupane State University, Bulawayo, Zimbabwe, 5 Faculty of Medicine, University of Khartoum, Khartoum, Sudan, 6 Suba University Hospital, Khartoum, Sudan, 7 Ibn Sina University, Khartoum, Sudan, 8 Centre for Tropical Medicine and Global Health, Nuffield Department of Medicine, University of Oxford, Oxford, United Kingdom

* wael.osman@ku.ac.ae

**Data Availability Statement:** All relevant data are within the manuscript and its Supporting Information files.

## Abstract

### Background

Asylum seekers, migrants, and refugees from African countries may have significant health needs, resulting in economic implications for receiving countries around the world. The risk of mental illness is higher in these communities because of factors like violence, deprivation, and post-immigration challenges.

### Objective

The purpose of this study was to examine the literature to determine the prevalence, predictors, and economic impacts of mental health (MH) disorders among asylum seekers, migrants, and refugees from African countries.

### Design and methods

In this scoping review, we followed the guidelines from PRISMA and CoCoPop. A modified version of the Appraisal Tool for Cross-Sectional Studies (AXIS) was used to assess study quality for cross-sectional studies, while an appraisal list was used for qualitative studies based on the Critical Appraisal Skills Programme (CASP). Inclusion criteria included peer-reviewed articles published in English, and articles based on official reports from credible institutions and organizations. Among the exclusion criteria were publications that were not peer reviewed or had not been sourced by credible sources, publications that did not meet the study topic or language criteria, mixed populations (including Africans and non-Africans), and research abstracts, reviews, news articles, commentary on study protocols, case reports, letters, and guidelines.

**Funding:** The author(s) received no specific funding for this work.

**Competing interests:** The authors have declared that no competing interests exist.

## Data sources

A systematic search was carried out in Medline (via PubMed), EMBASE, APA PsycINFO, Web of Science and EBSCO, to identify relevant articles that were published between 1 January 2000 and 31 January 2024.

## Results

A total of 38 studies met the inclusion criteria, including 22 from African countries and three qualitative studies. In terms of number of countries contributing, Uganda was the largest (n = 7), followed by Italy (n = 4). The most studied conditions, using multiple diagnostic tools, were Post-Traumatic Stress Disorder (PTSD, n = 19) and depression (n = 17). These studies all revealed elevated rates of mental health disorders among these groups, and these were related to migration, refugee-related factors, and traumatic events. Most of these groups are dominated by young males. There is, however, a prominent presence of minors and women who have suffered a variety of forms of violence, in particular sexual violence. Furthermore, mental illnesses, such as PTSD and depression, are not only persistent, but can also be transmitted to children. In accordance with our inclusion criteria, our review found only one study that examined the economic impact of MH disorders in these groups, leaving a significant knowledge gap. According to this randomized controlled trial, intervention to reduce psychological impairment can help young people stay in school, improve their quality-adjusted life year (QALY), and earn an incremental cost-effectiveness ratio (ICER) of $7260 for each QALY gained.

## Conclusion

Asylum seekers, migrants, and refugees from African countries are likely to experience MH needs, according to this scoping review. As well as posing persistent challenges, these disorders can also be transmissible to offspring. In addition to longitudinal studies of these groups, economic impact studies of mental illnesses are necessary.

## 1. Introduction

Human mobility is one of the most significant issues of the 21st century. The number of displaced persons worldwide is estimated to be the highest in history, with tens of millions crossing borders every day and approximately three billion crossings annually [1]. The number of forced migrants entering industrialized countries at an increasing rate in recent years has made screening health conditions more challenging [2].

Depending on the context, refugees, migrants, or asylum seekers may have a different meaning: political, media, security, health, or legal. According to the International Organization of Migration (IOM) [3], refugees are threatened because of their race, religion, nationality, membership in a particular social group, or political beliefs, and their own governments do not provide them with protection. "Migrants" are people who move across international borders or within states away from where they live regularly. A migrant may be a student abroad, a worker overseas, or an individual who comes to the country for economic reasons. Migrants and refugees differ primarily in how they experience forced migration. Migrants who migrate for economic, environmental, or family reasons are considered voluntary migrants. Asylum

seekers are individuals seeking international protection. In a country with individualized procedures, an asylum seeker is someone who has not yet been granted asylum by the country to which the claim has been submitted. It is important to note that asylum seekers are not all recognized as refugees, but that all refugees were originally asylum seekers. Detailed explanations of these terms can be found in the S1 File.

There has been an increase in refugee influxes into and out of African countries, creating the need for global responses to accommodate these populations. There have been reports of humanitarian concerns regarding the integration of the millions of people forced into exile by these flows into the social, economic, and political environments of their host countries. The result is that Africa continues to produce a disproportionate number of refugees; around 47 percent of the refugees registered with the United Nations High Commission for Refugees (UNHCR) come from the continent [4]. Therefore, it is difficult for host states to achieve a balance between maintaining control over their borders and protecting refugees.

Furthermore, migrants fleeing persecution of political opponents, human rights violations, and ethnic cleansing campaigns may have far-reaching effects. The number of refugees in Africa is quite high; Sudan, Mozambique, and Rwanda have produced more than a million refugees each [4]. When political turmoil occurs at home, it can have a significant spillover effect on other states and lead to strong reactions. Due to the lack of borders among refugees and dissidents, internal violence can easily cross national borders [5, 6].

The reactions of host countries to refugees are controversial and poorly understood. It has been reported that a significant number of refugees generate positive spillovers in host countries [7, 8]. In countries that have accepted refugees, the creation of new businesses and the replacement of aging populations have led to increases in income and gross domestic product [9]. However, despite these developments, it is very difficult to provide optimum benefits to refugees and asylum seekers due to resource limitations and the increasing level of global debt [10].

There may be barriers that asylum seekers, migrants, and refugee groups experience in coping with stress, working productively and fruitfully, and contributing to society. In addition to economic, sociocultural, and ecological factors, there are many factors and challenges contributing to poor mental health in these groups. Among them are language barriers, family separations, loneliness, documentation challenges, housing problems, loss of social standing, job insecurity, racism, and discrimination. Moreover, refugees who have experienced violent trauma or severe deprivation may be at greater risk of mental illness. According to a scoping review of 36 studies conducted in 12 African countries [11], lifetime prevalence of mood disorders ranged from 3.3 to 9.8%, anxiety disorders ranged from 5.7 to 15.8%, substance use disorders ranged from 3.7 to 13.3%, and psychotic disorders ranged from 1.0 to 4.4%. It is common for immigrants to experience stress as they leave their home country and enter a society that has different cultural values and social norms.

In terms of mental health (MH), refugees resettled in industrialized countries have encountered a variety of adverse experiences, including witnessing atrocities, losing family members, experiencing stressful escape and transit experiences, living in refugee camps, cultural and linguistic barriers, stigma surrounding mental illness, unemployment, financial hardship, and losing status, culture, and identity. Even so, most asylum-seekers are not screened for mental health problems during the asylum process [12], which is also true of most newly resettled refugees [13, 14]. As a result of this, forced migrant populations are known to have high rates of mental disorders, particularly major depressive disorder (MDD) and post-traumatic stress disorder (PTSD), which are several times higher than in the host population [15] or non-forced migrant populations [16]. Based on a large meta-analysis of refugees and other conflict-affected persons, the adjusted weighted prevalence rates for MDD and PTSD were 30% [17],

suggesting that refugees and asylum seekers are more likely to suffer from these mental disorders [18, 19], with asylum seekers experiencing even higher rates.

There is a critical need to promote mental health among asylum seekers, migrants, and refugees. While the circumstances and experiences of refugees and asylum seekers may differ, they all fled their countries of origin due to persecution, conflict, violence, or other dangerous circumstances [20]. During the resettlement process, migrants often encounter adverse circumstances and insecurity in their countries of origin, as well as challenges associated with the migration journey.

In light of this, this scoping review explored the literature to investigate the prevalence, predictors, and economic impacts of mental health disorders among asylum seekers, migrants, and refugees from African countries.

## 2. Methods

### The protocol, inclusion criteria, exclusion criteria

A study protocol based on Preferred Reporting Items for Systematic Reviews and Meta-analysis Protocols (PRISMA) was drafted for an MSc dissertation project at Liverpool Johns Moore University (LJMU), UK, and revised by the faculty board. As of now, the protocol cannot be accessed by the public, but it can be shared upon request.

Inclusion and exclusion criteria were established using the Context-Context-Population Framework (CoCoPop) in order to define prevalence data in accordance with the research question [21].

Articles considered in this review met the following inclusion criteria: (a) investigated the prevalence, predictors, and economic cost of mental illness and/or psychiatric disorders among refugees, migrants, and asylum seekers from African countries (definitions of all terms can be found in the S1 File) (b) were published in the English language, (c) were peer-reviewed and based on official reports from credible institutions and organizations and were published between 1 January 2000 and 31 January 2024.

The exclusion criteria included the lack of peer review or not coming from credible official sources and the inclusion criteria not meeting the study topic and language (English). The review excluded publications in which mixed populations were involved (both from Africa and elsewhere), in addition to research abstracts, reviews, news, commentary on protocol development, case reports, letters, guidelines, and books.

### Search strategy

We developed a comprehensive search strategy following the JBI Scoping Review Network guidelines. In order to gather all possible evidence in this field, an initial search was conducted for articles published between January 1, 2000, and January 31, 2024, including Medline (through PubMed), EMBASE, APA PsycINFO, Web of Science, EBSCO, Scopus and ProQuest. Additionally, we searched grey literature, such as the ProQuest theses database, and reviewed the references of eligible articles. We used a combination of Medical Subject Headings/Entree terms as well as free terms according to population, intervention, and outcome frameworks. S2 File provides detail of search terms.

### Screening, quality of evidence, and critical appraisal of studies

Following the search of the databases, and removal of duplicate articles, a total of 2475 articles were exported to Excel spreadsheet for further analysis. To screen the data, the first step was to review the abstracts for clear points that addressed the research question, screening full text if

the abstract was either unavailable or did not provide sufficient information. Exclusion of the papers was due to the following reasons: ineligible study types or populations, ineligible research questions, review articles, reports, or perspectives, no additional data, or duplication of publications. Additionally, we read references in papers to identify potential articles. We then categorized the searched papers into yes, no, and unclear categories, and included uncertain abstracts until we reviewed the full text articles to prepare the final list of eligible articles that were imported into EndNote (version X9) according to their eligibility. To assess the quality of cross-sectional studies, we used a modified version of the Appraisal Tool for Cross-Sectional Studies (AXIS) [22]. This tool is designed to ensure systematic interpretation and to inform decisions about the quality of cross-sectional studies. This critical appraisal (CA) method is usually used in systematic reviews, but we found it very useful in this scoping review study. The main components of this CA include assessing whether the study is suitable to answer the hypothesized question and whether bias might be introduced. According to AXIS tool, we used the following parameters for CA of studies:

1. The introduction: should include clear objectives.

2. Methods: a suitable study design; a sufficient sample size, usually > 100 except for quantitative studies; a target population (African); a clear definition of the condition to be studied, a validated instrument to assess it, and an appropriate statistical analysis plan for the described data.

3. Results: the basic data were adequately presented; respondents and non-respondents were described along with their reasons for not responding, and the results presented were described in the methods section.

4. Discussion: The authors' discussion and conclusions were supported by study results, and limitations of the study were addressed clearly.

5. Other: funding source explained appropriately; no conflict of interest; ethics statement available as needed.

For qualitative studies, a ten-point appraisal list consistent with the Critical Appraisal Skills Programme (CASP) tool was used. The CAPS tool covers key aspects such as study design, methodology, data analysis, and interpretation of results.

## Data charting

The key information we charted for all publications was the following: the author(s), publication year, origin/country of the study (where it was conducted), category of the population whether it is refugees, immigrants, or asylum seekers, study population and sample size (if applicable), age categories, gender percentages, techniques, approaches, and methods used to assess the MH under study, prevalence of the MH problem(s) under study, predictors and determinants of the prevalence of the MH problems under study, and an overview of key findings related to one or more of the scoping review questions. For the economic burden studies, we extracted data about the approach used for the economic evaluation of the studied MH problem. We also extracted data about study type and design, and the study outcome.

## Data quality assurance and appraisal of individual sources

To achieve this goal, we used the condition-context-population-framework (CoCoPop) to determine data eligibility. In addition to the comprehensive search which included searching the grey literature, particularly theses through the ProQuest database, we included only articles

which were written in English to avoid misinterpretations and considered both the rationale and limitations of publications. The data charting process included reviewing evidence from all sources and using predefined charting formats. Following the inclusion of studies and the recording of key information relevant to review questions, we developed a proposal to present results in accordance with the objectives or review questions using diagrams, tables, or descriptive formats.

### Data analysis & synthesis

The extracted data were compiled using the narrative synthesis of Barnett-Page and Thomas [23], and we followed the guidelines for data synthesis without meta-analysis [24].

## 3. Results

### Characteristics of the included studies

**Fig 1** summarizes the results of the study, which included 38 articles that met the inclusion criteria.

Based on the 38 studies included in this review, 35 were quantitative cross-sectional studies, while three were qualitative exploratory studies. A summary of the important findings of the eligible articles is presented in **Table 1** and **Fig 2** illustrates the country location of each study. As shown in **Fig 2**, the majority of the studies were conducted within the African continent, with 22 studies being conducted on the continent. Among the studies, seven were from Uganda, followed by four from Italy, and three from the United States. In most of these studies, both males and females are examined, focusing on refugees' adult populations. A single study focused on women, two on unaccompanied minors, one on children and parents as dyads, and one on young adults. In the select studies, gender representation varies, but most studies include more males than females. According to most studies, participants were less than 40 years old on average. East Africa represented the majority of study participants, followed by West Africa. No study included populations from Northern or Southern Africa. A study of unaccompanied Eritrean minors in Sudan [25], used both research approaches. In this study, after researchers determined the prevalence of anxiety and depression in the group, they used a quantitative approach to develop themes based on the daily stressors associated with these mental health disorders.

### Description of the major conditions and the psychometric instruments used in the examination

A wide range of mental and psychiatric conditions have been examined in the studies reviewed in this article. Across all studies, PSTD or CPSTD was the most frequently studied condition. Study focus, however, may vary from assessing a specific condition such as PSTD or depression to assessing the overall mental health of the study population, which may include anxiety, depression, and PSTD together. **Fig 3** summarizes the number of studies that examined different mental disorders. It should be noted that each study examined more than one condition, thus the number of studies does not equal 38 when taken as the total number of all studies included.

Studies often used more than one set of instruments to assess the prevalence and patterns of a particular condition, such that different instruments were used for different conditions. An overview of the main instruments used to assess the different conditions is provided in **Table 2**.

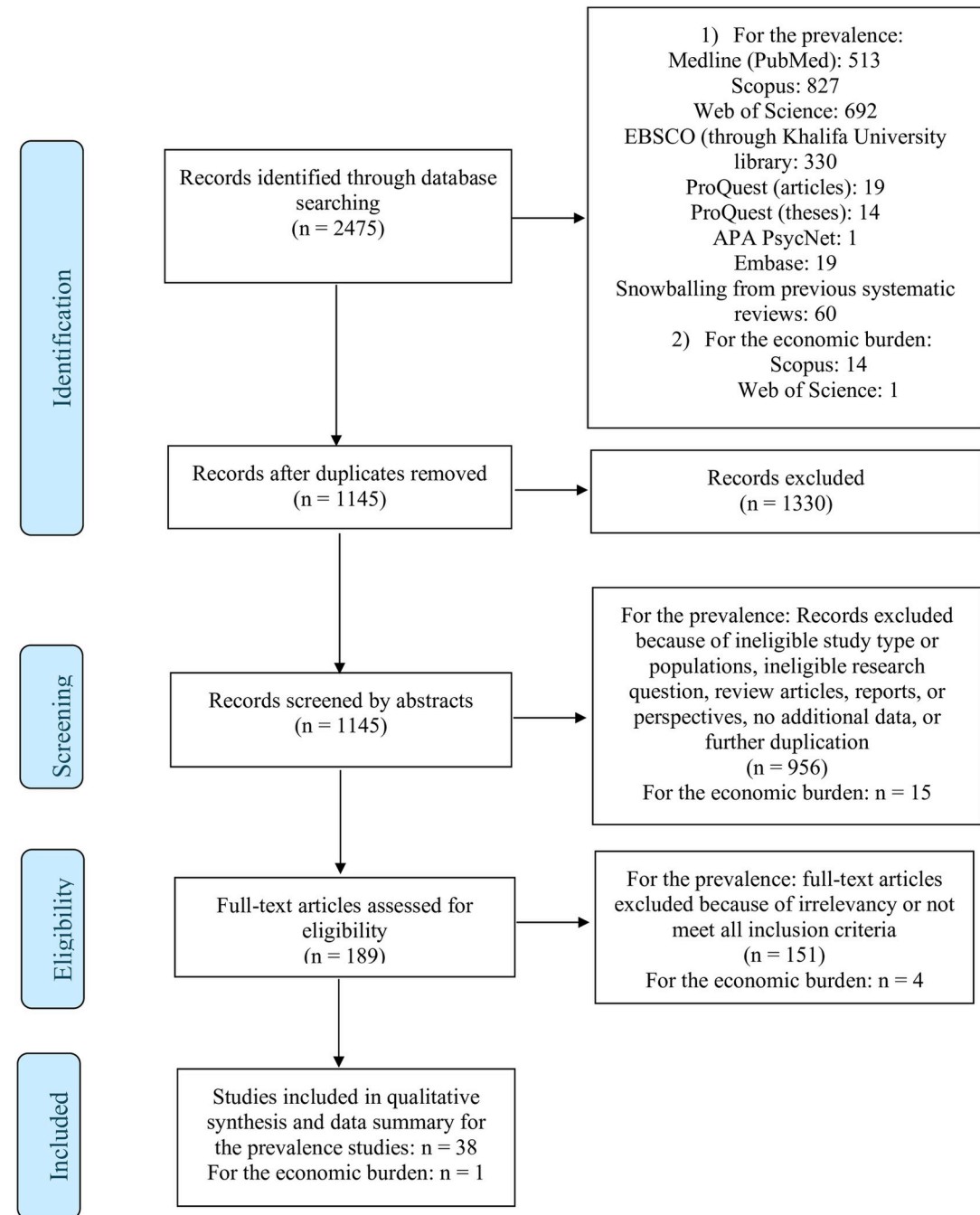

**Fig 1. Flow chart illustrating the search and identification of eligible articles according to the PRISMA chart style.**

## The prevalence of MH disorders among African asylum seekers, migrants, and refugees

It is evident from **Table 2** that mental health disorders have a high prevalence in all studies, varying according to the demographics of the study population, the specific mental health condition, and the location of the study population. Although it is difficult to compare prevalence across these factors since different instruments are used for evaluating prevalence, as well as

**Table 1. Findings of the study based on population, host countries, sex and age distributions, prevalence, and diagnostic indicators, as well as risk predictors.**

| Ref Study | Population | Place of the study | Population category | Category§ | Sample size | Age (M and/or SD or range) | Female (%) | Studied condition | Study instruments§ | Prevalence of studied condition (%) | Predictors of the prevalence |
|---|---|---|---|---|---|---|---|---|---|---|---|
| [26] | Congolese | Uganda | Adults | R | 325 | 30.9 (FM), 31.9 (M) | 56% | PSTD | PSS-I | Women: 94% Men: 84% | Severe trauma, rape in females |
| [27] | Eritrea and Sudan | Israel | Adults | A | 355 | 35.15 ± 8.29 | 31.80% | Suicidal ideation | HTQ, PHQ-9 | 31% | Post-migration living difficulties |
| [28] | Liberians, Sierra-Leoneans, and Togolese | Nigeria | Adults | R | 444 | 34.8 ± 12.8 | 40.8 | Mental Health | MINI, WHOQOL-BREF and CQoL | Depression: 45.3% Obsession: 34% PSTD: 34% Mania: 25.9 Suicidal ideation: 11%. | Unskilled workers, unemployed |
| [29] | West African (Fulani population) | USA | Parent-child dyads | M | 91 | 39.3 ± 8.7 for parents, 8.18 ± 2.04) for the children | 61% (parent), 39% (children) | Parental Trauma and child externalizing behavior | HTQ for the PSTD, CBCL Externalizing for child externalizing behavior. A 4-item self-report scale assessed difficulty parenting in the last month. | Children of immigrants recovering from trauma are at risk of exhibiting behavioral symptoms | Parents' HTQ scores, parenting difficulty, length of parent-child separation, and child age |
| [30] | West Africans | Italy | Adults | A | 385 | 23 (20–27) | 9% | Mental Health | Mental health screening, unspecified | PSTD: 31% Depression: 20% | Being in a combat situation or at risk of death, having witnessed violence or death and having been in detention |
| [31] | Multiple (mainly Sudan) | Niger | Adults | A | 126 | 26.12 ± 6.88 | 24.60% | PTSD and CPTSD | ITQ | CPTSD: 74.6% PTSD: 19.8% | Early age of the trauma, inadequate reception conditions in large, isolated facilities in the host country |
| [32] | Multiple | Italy | Adults | A & R | 122 | 25.11 ± 6.66 | 13.90% | PSTD | HTQ, PTSD Checklist for DSM-5, and LCA | 79.50% | Living in large reception centers |
| [33] | Multiple (mostly West Africans) | Italy | Adults | R | 120 | 25.1 ± 6.7 | 14% | PTSD and CPTSD | PSTD: HTQ, PDS, and PCL-5) CPTSD: ITQ | CPTSD: 30% PTSD: 38% to 79%. | Months spent as a refugee |
| [34] | East Africans (predominantly Somali) | USA | Adults | R | 52 | 37 ± 20.91 | 27.70% | Postmigration living difficulties (PMLD) | Depression: HSCL-25. PSTD: HTQ. Postmigration living difficulties: PMLD | The PMLD most frequently endorsed as a moderately serious to very serious problem | Duration of residency at the USA |
| [35] | Somali | UK | Adults | R | 180 | 40.4 (20–88) | 49.40% | Psychosis (BPRS), anxiety and depression (SCL–90) and suicidal thinking (BDI) | HSCL-25, psychiatric symptoms: HTQ and Brief Psychiatric Rating Schedule. | - | Shortages of food, being lost in a war situation, and being close to death and suffering serious injury |

*(Continued)*

**Table 1.** (Continued)

| Ref Study | Population | Place of the study | Population category | Category[¶] | Sample size | Age (M and/or SD or range) | Female (%) | Studied condition | Study instruments[§] | Prevalence of studied condition (%) | Predictors of the prevalence |
|---|---|---|---|---|---|---|---|---|---|---|---|
| [36] | Somali | Ethiopia | Adults | R | 847 | 33 (18–40) | 53.90% | Depression | HTQ and PHQ-9 | 38.30% | Gender, marital status, displaced previously as refugee, witnessing murderer of family or friend, lack of house or shelter and being exposed to increased number of cumulative traumatic events |
| [37] | West Africans | Senegal and Ivory Coast | Young adults | R | 123 | | 52% | Mental Health | Qualitative study by semi-structure interviews with young people and health professionals. | 8% | Preimmigration and immigration condition, difficult living conditions, long asylum procedure |
| [38] | Somali | Kenya | Women | R | 209 | 29 | 100% | Mental Health due to gender-based violence (GBV) | Anxiety: GAD-7 scale. Depression: PHQ-9. PTSD: HTQ | Anxiety:41% Depression: 36% PTSD: 3% | Intimate partner violence or conflict-related violence |
| [39] | Zimbabweans | South Africa | Adults | R | 125 | 18–48 | 42.40% | PTSD | GHQ-28 and the PTSD-PCL | 21% to 62% | For women: threat to life and rape before age 18. For men: death of a family member and forced sex after age 18. |
| [40] | Somali and Ethiopian (Oromo) | USA | Adults | R | 1134 | 35.1 ± 13.9 | 46.60% | Psychological trauma due to torture | Clinical Assessment and PCL-C | 25% to 69% | Torture |
| [41] | Sudanese | Uganda | Adults | R | 1,240 | 29.7 ± 9.6 | 77.9% | PSTD | PDS | 48% | Sex, age, education and occupation |
| [42] | Ugandan and Sudanese | South Sudan | Adults | R | 3339 | 15–50 | 75% | PSTD | PDS | 50.50% | Cumulative trauma and avoidance |
| [43] | Liberian | Nigeria | Adults | R | 520 | 33.04 ± 10.9 | 35.80% | Psychological Distress | Depression, Anxiety, and Stress scale | Anxiety: 31.7% Depression: 67.9% Stress: 40.2% | Shame and lack of social support |
| [44] | Congolese | Uganda | adolescent | R | 89 | 21.08 ± 1.98 | 62.90% | PSTD | PDS | PTSD: 49.4% | Trauma load |
| [45] | Sub-Saharan Africans | Sweden | Adults | R | 420 | 33 (16–80) | 47.90% | Post-Migration Stress | HTQ, PMLD, the Cultural Lifestyle Questionnaire; and the HSCL-25 | PSTD: 47% Depression: 20% | Shorter duration in Sweden, greater number of traumatic events |
| [46] | Congolese | Kenya | Adults | R | 245 | 18–80 | 50.60% | Psychological Distress | SRQ-20 | 52.80% | Perceiving to have a useful role in one's family or community, feeling afraid to leave home for medical/ health care, ethnic Banyamulenge Congolese adults without legal refugee |

(*Continued*)

**Table 1.** (Continued)

| Ref Study | Population | Place of the study | Population category | Category¶ | Sample size | Age (M and/or SD or range) | Female (%) | Studied condition | Study instruments§ | Prevalence of studied condition (%) | Predictors of the prevalence |
|---|---|---|---|---|---|---|---|---|---|---|---|
| [47] | Northern and Sub-Saharan Africans | Italy | Adults | M | 293 | 31 (27–40) | 28.30% | Depression, anxiety, PTSD | Depression: PHQ-2. Anxiety: GAD-2. PTSD: PC-PTSD-5. | Anxiety: 12.1% Depression: 21.1% PTSD: 24.4% | Discrimination and waiting to be in possession of temporary permits |
| [48] | Multiple | Hong Kong | Adults | M | 374 | 31.52 ±7.41 | 21.90% | Depression | PHQ-2 | 36.10% | Discrimination, and difficulties accessing medical services |
| [49] | Multiple (mostly Nigeria and Zimbabwe) | Australia | Adults | M | 167 | 25–54 | Not reported | Depression, Distress, and Coping strategies | Distress: K10 Depression: PHQ-9 Coping strategies: Brief COPE | Depression and psychological distress in younger age groups is 62.5%, and in the older age groups is 22.2% | Self-blame, self-distraction, and behavioral disengagement, religious coping |
| [50] | Eritrea or Sudan. | Israel | Adults | A | 101 | 31.73 ± 7.81 | 29.7% | PTSD, Mood disorders, Psychotic Disorder, others | ICD-10 | PTSD: 51.5% Mood disorders: 10.9% Psychotic Disorder: 10.9% Substance abuse: 8.9% Suicide attempts: 6.9% | Authors did not report the predictors of each condition but the predictors of the dropout from the therapy which were: years in Israel, family status, and those in the psychotic spectrum disorder |
| [51] | Sub-Saharan Africa and the Eastern Mediterranean Region | Morocco | Adults | A, M, R | 445 | 27.9 ± 10.9 | 31.2% | Anxiety and Depression | HADS-25 | Anxiety: 39.1% Depression: 40.0% | Anxiety: Diabetes, refugee status, overcrowding in the home, stress, age between 18 and 20 years, and low income. Depression: lack of social support and low income. |
| [52] | East Africans | Kenya, Uganda and Ethiopia. | Adults | R | 15, 915 in total (8303 refugees + 7612 host national | 30.08 ± 11.32 | 51.0% | Depression and Functional impairment | Depression: PHQ-9 WHODAS 2.0 | Refugees: Depression: 31.0% Functional impairment: 62.0% Host nationals: Depression: 10.0% Functional impairment: 25.0% | High level of violence |
| [26] | Congolese | Uganda | Adult | R | 325 | FM (30.9) M (31.9) | 56% | PSTD (war-related) | PSS-I and the 17 DSM-IV symptoms criteria for PTSD | All cohort: 89% Women: 93.8% Men: 83.7% | For all cohort: severe trauma (e.g. rape, witnessing murder,) For women: rape |

(*Continued*)

**Table 1.** (Continued)

| Ref Study | Population | Place of the study | Population category | Category¶ | Sample size | Age (M and/or SD or range) | Female (%) | Studied condition | Study instruments§ | Prevalence of studied condition (%) | Predictors of the prevalence |
|---|---|---|---|---|---|---|---|---|---|---|---|
| [25] | Eritrean | Sudan | Unaccompanied minors (URM) | R | 45 | 15.36 (1.45) | 37.8% | Depression and anxiety, | HSCL-25 and | 88.9% for both depression and anxiety | Younger age for both conditions, foster care duration for anxiety |
| | | | | | | | | Qualitative analysis | Thematic analysis through interview | Broad themes of daily stressors | Identified themes are harassment (verbal, sexual, and physical), racism, financial difficulties, Acculturation Distress, and Restricted freedom |
| [53] | Congolese Swahili and Somalians | Uganda | Adult females | R | 117 | 31.6 | 100% | PSTD and depression | HSCL-25, HTQ | Depression: 92.0% PSTD: 71.1% | Gender-based violence (GBV): physical and sexual |
| [54] | 9 countries (mostly eastern Africans) | Uganda | Adult | R | 387 | 33.01 (12.2) | 56.59% | Psychiatric disorders | MINI | Anxiety: 73% PTSD:67% Depression: 58% Substance: 30% | Having PSTD |
| [55] | Sub-Saharan Africans | China | Adult | M | 928 | 26 ± 8.7 | 38% | Depression | CES-D | 44% | Unsatisfactory housing conditions, perception of very unfriendly attitudes from the local people |
| [56] | Eritrean | Ethiopia | Adults | R | 786 | 30 (median) | 37% | Depression | PHQ-9 | 37.8% | Age (older age), sex (female), educational status (never), occupation (unemployed), displacement history, poor social support, personal and family history of psychiatric illness, current substance use, length of stay at the refugee camp, traumatic event and current presence of family |
| [57] | Eritrean | Ethiopia | Adults | R | 396 | 18–60 | 48.5% | Suicide Attempts | Interviews and the WHO survey questions: Having you ever attempted suicide?", (definition: A suicide attempt was defined as potentially self-injurious behavior with a non-fatal outcome, where the person intended at some level to kill themselves) | 7.3% | Trauma, family history of mental Disorder, personal history of PSTD, severe hopelessness |

*(Continued)*

**Table 1.** (Continued)

| Ref Study | Population | Place of the study | Population category | Category[¶] | Sample size | Age (M and/or SD or range) | Female (%) | Studied condition | Study instruments[§] | Prevalence of studied condition (%) | Predictors of the prevalence |
|---|---|---|---|---|---|---|---|---|---|---|---|
| [58] | Burundian | Tanzania | Families | R | Children (n = 230) and their parents (n = 460) | Children: 12.11 (2.03) Mothers: 34.49 (8.48) Fathers: 41.52 (11.00) | Children: 47.4% | Suicidality (ideation, plans, and attempts) | MINI and MINI-KID | Ideation, plans, attempts, respectively Children 11.3%, 0.9%, 0.9% Mothers: 37.4%, 7.4%, 5.2% Fathers: 29.6%, 4.8%, 1.7% | Children: older age, PSTD, externalizing problems Mothers: violence, living in larger households, higher psychological distress Fathers: war-related trauma |
| [59] | Ethiopian | Canada (Toronto) | Adults | M and R | 342 | 35.3 | 46.6% | Depression | CIDI | 9.8% | Trauma, camp internment, postmigration stressful life events, and discrimination |
| [60] | Five African countries | Uganda | Adults | R | 343 | 42.3% | M: 28.3 (11.2) F: 29.5 (10.5) | substance use | AUDIT–10) and DAST–20 | Alcohol use: 43% Other substance use: | Alcohol use: depression and being separated from spouse. Other substance use: higher age and being male |
| [61] | Many African countries (mostly Gambia, Somalia, and Nigeria) | Austria | Unaccompanied minors (URM) | R (A) | 41 | 14.6% | 16.95 (0.82) | PTSD | UCLA PTSD Reaction Index, and Scales for Children Afflicted by War and Persecution (SCWP) | PSTD (full criteria): 17% PSTD (partial criteria): 29.3% | Being a girl and war-related traumatic events |

[¶] In the category column, R indicates refugees, A indicates asylum seekers, and M indicates migrants.

[§] Please refer to Table 2 for the abbreviations for the instruments used for determining prevalence.

Abbreviations in the age column: M: mean, SD: standard deviation.

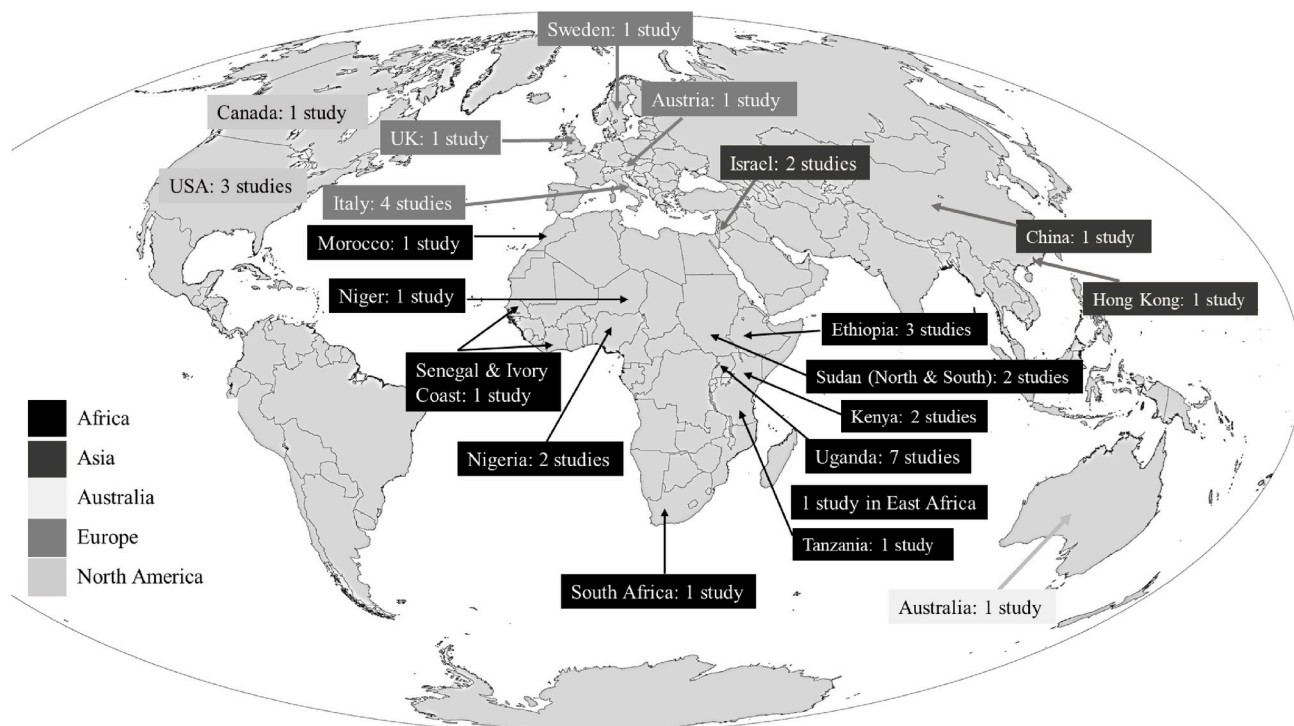

**Fig 2. Distribution of the 38 eligible studies across the world.** The world map in the figure was taken from NASA (https://eoimages.gsfc.nasa.gov/images/imagerecords/47000/47427/global_tm5_mangroves_lrg.png).

the different contexts and conditions assessed in each study, certain patterns are evident. Firstly, women have a higher prevalence of specific mental health conditions. According to a study conducted on Congolese populations in Uganda [26], 94% of women reported symptoms of PTSD, compared to 84% of men [26]. A similar observation was made in another study examining female Congolese and Somali populations in Uganda, where 92% and 71.1% of men and women reported having PTSD and depression, respectively. Additionally, research conducted within African countries tends to show a greater prevalence of different conditions

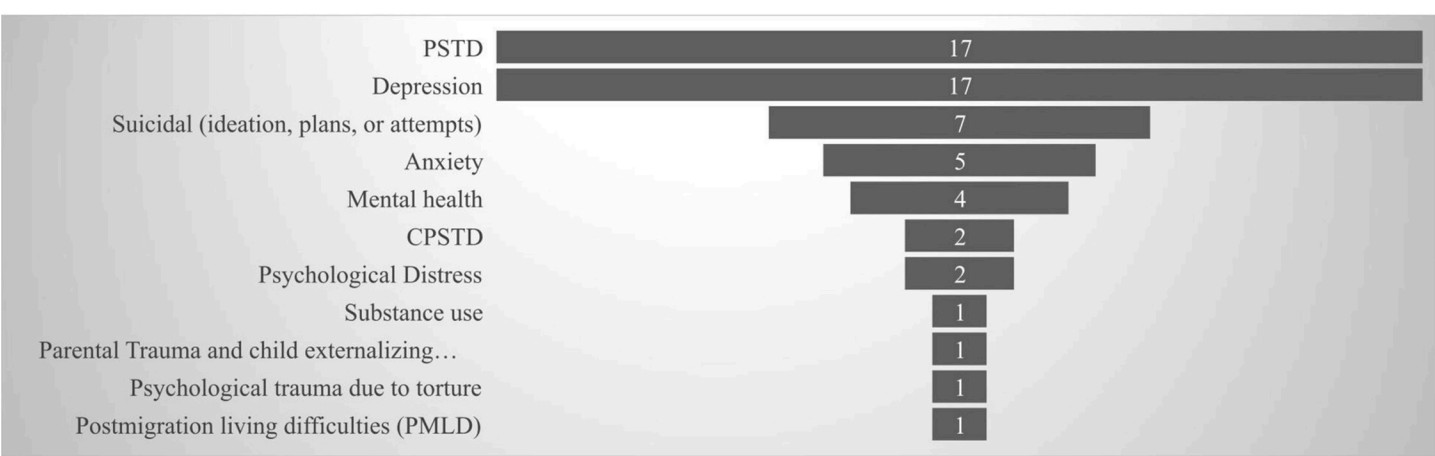

**Fig 3. Number of studies conducted for each mental disorder condition examined.**

than research conducted elsewhere in the world. It is difficult, however, to draw any definitive conclusions based on the different tools used.

Second, higher prevalence is associated with specific mental health conditions. There is a high prevalence of PSTDs and CPSTDs across the included studies (range: 19.8% to 94%), perhaps due to the fact that they are the most studied conditions [26, 31]. There was a similar trend in the 17 studies that examined the prevalence of depression. Most studies reported a prevalence of less than 50%, with a range between 20 and 67.9%. In other studies, similar trends have been observed, however, suicidal ideation (7.3–31%) has been reported to be the least common trend (which is not high when viewed in conjunction with the prevalence of conditions associated with suicidal ideation, including depression). These data indicate that asylum seekers, migrants, and refugees' groups from African countries suffer from higher rates of mental illness, specifically PTSD and depression, than the general population.

## Qualitative studies overview

A qualitative examination of specific mental health conditions was conducted in three of the studies in this review. Anakwenze and Rasmussen [29], investigated the effect of parental trauma on the externalizing behavior of children using parent-child dyads. In another study by Gakuba and colleagues s [37], examined the mental health of young adult refugees, as well as the factors that predispose this population to poor mental health (**Table 2**). As part of their study, a semi-structured interview process conducted with youth and professionals working in the mental health field among asylum seekers, migrants, and refugees, themes for reasons for poor mental health among these populations were identified, including immigration, and living conditions, as well as the lengthy asylum process.

## Results of the study in children and unaccompanied minors

Children were involved in five studies, either as unaccompanied minors (two studies) or adolescents (one study) or as participants in studies in which one or both parents took part (two studies). There is a marked difference in the prevalence of mental health problems among unaccompanied minors in Africa and Europe; Eritrean minors in Sudan showed 88.9% anxiety and depression, and African minors in Austria demonstrated 17 to 29% depression [25, 61]. One interesting study is that by Scharpf and colleagues [58] which shows that when examining the whole Burundian families for suicidality, mothers and fathers showed different discrepancies, with mothers being most affected than fathers, and children exhibiting lower rates when living with their parents. According to these findings, suicide is predicted differently by family members, with children more likely to commit suicide if they suffer from externalizing disorders, mothers mostly due to violence, and fathers due to war-related trauma.

## Predictors of different conditions' prevalence

The studies that were included provided snapshot prevalence data, which cannot be used for developing a mathematical model that can be used for predicting the prevalence of the disease. The term "predictors" is therefore used to describe the factors and conditions associated with the occurrence of a variety of mental health disorders, rather than to refer to predictive models or factors that can be used to predict their prevalence.

There are certain demographic characteristics that are associated with a higher prevalence of MH disorders and depression, namely older age, female gender, having no formal education, being unemployed, and living in poor housing conditions [11]. Further, it has been found that displacement history, poor social support, separation of the family during migration, post-migration isolation, and hostile attitudes from host populations are all important

**Table 2. A summary of the key diagnostic instruments used in all 38 studies along with their abbreviations.**

| Condition assessed | Instrument (scale) | A brief description of the instrument |
|---|---|---|
| PSTD | The Harvard Trauma Questionnaire (HTQ) | 47 items describing traumas that occurred before migration + 16 items assessing PTSD symptoms. |
| | PTSD Symptom Scale–Interview (PSS-I) | An interview consisting of 24 semi-structured items designed to assess PTSD symptoms over the past month. |
| | PTSD Checklist for DSM-5 (PCL-5) | Self-report measure that assesses 20 symptoms of PTSD listed in the DSM-5. |
| | The 17 DSM-IV symptoms criteria for PTSD | 17 DSM-IV symptoms related to PTSD, which are assessed using one question for each symptom. |
| | UCLA PTSD Reaction Index | A semi structured interview evaluates a child's exposure to 26 types of traumatic events and assesses their past-month frequency of PSTD symptoms (from "none of the time" to "most of the time") using 48 items. A Parent/Caregiver interview is also available, with both a general version and a version for children under 6 years old. |
| | Post-traumatic Stress Diagnostic Scale (PDS) | A self-administered test takes 10 to 15 minutes and requires a reading age of 13 years. As well as a total severity score, it provides subscale scores for intrusions, avoidance, and arousal symptoms. |
| | The International Trauma Questionnaire (ITQ) | An ICD-11-based 12-item self-report measure that assesses the core symptoms of PTSD and CPTSD |
| PTSD and CPTSD | The Beck Anxiety Inventory (BAI) | Assessment of anxiety symptoms using a 21-item self-administered questionnaire |
| Anxiety | The Generalized Anxiety Disorder 7 (GAD-7) scale | Seven questions on which cut-off scores of 5, 10 and 15 are recommended for mild, moderate, and severe anxiety, respectively |
| Generalized anxiety disorder | The Hopkins-25 Checklist (HSCL-25) | Ten items for anxiety and 15 items for depression. |
| Anxiety and depression | The Depression Self- Rating Scale (DSRS) | An 18-item self-report instrument for assessing juvenile depression symptoms among children and adolescents. |
| Depression | The Centre for Epidemiologic Studies Depression Scale (CES-D) | An evaluation of caregivers' depression symptoms. Scores range from 0 to 60, with high scores indicating greater depression symptoms. Cutoff scores (16 or greater) provide good sensitivity, specificity, and internal consistency in identifying individuals at risk for clinical depression. |
| | The WHO Composite International Diagnostic Interview (CIDI) | It is a comprehensive and fully standardized diagnostic interview that measures mental disorders based on ICD-10 and DSM-III-R diagnostic criteria. A total of 276 symptoms are asked, many of which are connected to probe questions to evaluate severity, as well as questions assessing help-seeking behavior, psychosocial impairments, and other episodes. |
| | The Patient Health Questionnaire-2 (PHQ-2) | A two-item depression screening instrument. |
| | The Brief Patient Health Questionnaire (PHQ-9) | A self-administered nine-item questionnaire assessing depression symptoms over the past two weeks. |
| Post-Migration Living Difficulties | The Post-Migration Living Difficulties (PMLD) questionnaire | 25 questions assess financial, healthcare, family, discrimination, and immigration difficulties. |
| Mental Health | The Mini-International Neuropsychiatric Interview (MINI). | A structured diagnostic interview that evaluates psychiatric patients' diagnoses by asking 26 questions with "Yes" or "No" responses in less time than other diagnostic interviews. |
| | MINI-KID | Mini-International Neuropsychiatric Interview for Children and Adolescents (MINI-KID) is a short, structured diagnostic interview for children and adolescents 6–17 years of age with psychiatric and suicidal disorders as defined by DSM-IV and ICD-10. In his standard version, 30 disorders or subtypes of disorders in pediatric mental health are assessed. |
| Substance use | The Alcohol Use Disorders Identification Test (AUDIT– 10) | A simple method developed by the WHO for screening for excessive drinking and for short assessments to help people identify potential alcohol-related problems. |
| | Drug Abuse Screening Test (DAST– 20) | Self-report or structured interview formats are available for this 20-item instrument. Each question can be answered "yes" or "no". This test is constructed similarly to the Michigan Alcoholism Screening Test (MAST), and the items are similar. This method identifies people who misuse psychoactive drugs and yields an index score of the degree of difficulties associated with drug abuse. |
| Quality of life | The Community QoL questionnaire | A twelve-question questionnaire that measures an individual's physical, psychological, and social well-being. |
| Parent report measure of child behavior | The Child Behavior Checklist (CBCL) | A widely used caregiver report form provides caregivers with 99 items to rate in terms of problem behavior in children |

(*Continued*)

**Table 2.** (Continued)

| Condition assessed | Instrument (scale) | A brief description of the instrument |
|---|---|---|
| War trauma | The War Trauma Screening Scale (WTSS) | A 72-item self-report checklist about violence and adversity experienced because of exposure to war |
| Discrimination | Everyday Discrimination (EDD) | An evaluator-administered, 9-item self-report instrument designed to measure ongoing, routine, and minor incidents of discrimination among young people |
| Distress | Kessler Psychological Distress Scale (K10) | The self-report questionnaire includes 10 items and is used to measure a person's overall distress based on anxiety and depression symptoms experienced in the previous 4 weeks. |
| Coping strategies | Brief COPE | Generally, "coping" refers to methods of minimizing distress caused by negative life experiences. This 28-item self-report questionnaire measures how well an individual copes with stressful life events by comparing effective and ineffective methods. The test can be used to assess someone's ability to cope with a wide range of difficulties. These difficulties include severe medical conditions, injuries, traumatic events, natural disasters, and financial strain. |

predictors that have been identified to explain the change in behavior [11]. A number of conditions indicated different predictors that could be attributed to migration or refugee factors, which are mainly traumatic events, as well as postmigration refugee factors, primarily the length of time spent as a refugee or the time spent waiting for asylum, the difficulty of getting medical care, as well as discrimination. Women are more likely to be affected by various forms of gender-based violence, such as sexual violence, rape, or interpersonal violence than men, which may explain why this review found a higher prevalence of PTSD in women across a number of studies.

In regards to refugee populations in host countries outside Africa, a qualitative study found that there are certain daily stressors that lead to mental health disorders and higher levels of psychosocial stress for these populations [25]. There were several themes identified, including harassment (verbal, sexual, and physical), racism, financial difficulties, acculturation distress, and restricted freedom. **Table 3** summarizes the major predictors of the various conditions across all the studies.

## Economic burden of mental health disorders among African asylum seekers, migrants, and refugees

Despite our comprehensive review of the economic ramifications of mental health disorders among African asylum seekers, migrants, and refugees, we identified only one study that

**Table 3. Summary of the major predictors and reasons of common mental health disorders in asylum seekers, migrants, and refugees' groups from African countries derived from 38 studies.**

| Condition | The major predictors across all studies |
|---|---|
| PSTD | • The severity of trauma<br>• Torture and rape (particularly in women)<br>• More time spent in camps or reception centers as a refugee |
| Depression | • Discrimination<br>• Gender (female)<br>• Waiting for the outcome of the case<br>• Unemployment |
| Anxiety | • Waiting to possess a temporary permit.<br>• Discrimination |
| Suicidal ideation | • Depression or family history of depression<br>• Post-migration living difficulties.<br>• Trauma and violence |
| Psychological distress | • Lack of social support, harassment, racism, financial difficulties |
| Substance use | • Alcohol use: depression<br>• Other substances: being male |

examined the specific economic impacts of mental health interventions [62]. The study used the Youth Readiness Intervention (YRI), a behavioral intervention designed to reduce functional impairment (expressed in terms of quality-adjusted life years - QALYs) among war-affected young people in Freetown, Sierra Leone who were not able to attend school (**Table 4**). In the secondary analysis, financial and economic costs were analyzed as well as incremental cost-effectiveness ratios (ICERs) based on gains across mental health and education dimensions. In the YRI group, school retention was higher, functional impairments were lower, and the ICER per QALY gained was $7260. According to the authors, when the willingness-to-pay threshold exceeds three times the average gross domestic product per capita, the YRI is not cost-effective. Despite this, the results of the YRI indicate that a range of benefits were derived, including an increase in enrollment in schools, which was not captured in the cost-effectiveness analysis.

## 4. Discussion

This study examined mental health disorders among asylum seekers, migrants, and refugees from African countries without any restrictions regarding the refugee experience (region of origin, duration of displacement, or country of settlement, among others). One important observation is that most asylum seekers, migrants, and refugees tend to relocate within Africa, whereas younger age groups tend to leave the continent, particularly to Europe. There may be a war-related explanation for this trend. For example, the Libyan war, which has left the country unable to control its borders. As a result, West Africa is cut off from the Mediterranean Sea, which provides access to Europe. Similarly, Sudan, with its long history of conflict and border with Libya, offered an alternative path to Eastern Africans.

It appears that most of the studies were conducted before 2020, suggesting that the COVID-19 epidemic may have negatively impacted refugee movements. Even so, it is expected that these disputes will continue to escalate due to ongoing conflicts and tensions in Africa, as well as the global economic crisis, as the Lampedusa refugee crisis in Italy recently

**Table 4. Study findings regarding the economic burden of MH for Sierra Leonean refugees.**

| Ref Study | Population | Place of the study | Demographic data | Category | Study design | Economic burden studied | Analysis approach | Key findings |
|---|---|---|---|---|---|---|---|---|
| [62] | Sierra Leone | Sierra Leone | Sample size: 436. Population type: Youth are not attending school. Age range: 15–24 Female %: 45.6% | R | Randomized controlled trial. Two groups of participants were randomized into the study: 222 received Youth Readiness Intervention (YRI) reducing functional impairment associated with psychological distress among war-affected young persons and 214 received usual care. Both groups received an education subsidy immediately following the intervention. | Cost-utility analyses (CUA) as a function of quality-adjusted life year (QALY). | The primary analysis used an 'ingredients approach' to estimate the financial and economic costs. A cost-effectiveness ratio (ICER) expressed in terms of mental health and schooling gains was also calculated. A secondary analysis examined whether intervention effects were greatest among the worst-off (upper quartile) psychological distress at baseline. | Primary analysis yielded an ICER of $7260 per QALY gained. Secondary analyses indicated that the intervention was cost-effective ⸙ among those worst-off psychological distress at baseline, yielding an ICER of $3564 per quality-adjusted life years (QALY) gained. However, observations were seen in until 6-month follow-up duration and not after that. |

⸙ Cost effectiveness was defined as meaning that each QALY gained would cost less than three times the country's average per capita gross domestic product (GDP) at purchasing power parity.

Abbreviations: R: refugees.

demonstrated. Currently, the British government is proposing a number of laws to control asylum seekers, migrants, and refugees in the UK, which illustrates how local governments in Europe are becoming more aware of the importance of future trends in asylum seekers, migrants, and refugees.

## Refugees, asylum seekers, and migrants from African countries are more likely to suffer from mental health disorders

Based on this research, African asylum seekers, migrants, and refugees suffer from high rates of PTSD (19.8% to 94%) and depression (20% to 67%). The prevalence of both conditions in African populations is significantly higher than in global migrant and refugee populations (PTSD prevalence = 31.5%; depression prevalence = 31.5%) [63]. One possible explanation is that most forced migrations within the continent are the result of war or violence. Our research shows that war survivors and refugees from conflict-affected regions suffer from higher PTSD rates [64, 65]. Nevertheless, African populations also have significantly higher prevalence rates than other war-affected migrant and refugee groups such as Syrians resettled in Western countries, who report prevalence of PTSD and depression at 31.5% [66]. The reason for this observation is unclear and warrants further research, however, we hypothesize that the poor living conditions of African refugees and migrants in host African countries and the challenges faced by racism and social integration in Western host countries may contribute to higher mental health conditions.

Moreover, the prevalence rates of PTSD and depression in African refugee and migrant populations are higher than those in the general population, where lifetime prevalence is estimated at 3.9% and 10.8%, respectively [67, 68]. In light of this evidence, the migration morbidity hypothesis [69] is supported and the healthy migrant hypothesis [70] may be contested. There are two possible explanations for this. First, African refugee and migrant populations in host African countries continue to live in poor conditions and suffer from unemployment, both of which contribute to the prevalence of morbidity. Secondly, it is notable that a significant portion of the population in these select studies has experienced forced migration rather than voluntary migration, which, according to the literature, is highly selective of the younger and healthier migrants.

Similarly, we establish that anxiety disorders and psychosis are less prevalent among African asylum seekers, migrants, and refugees. This is consistent with the results of systematic reviews of global refugee and migrant populations that indicate a lower prevalence of anxiety disorders and psychosis (11% and 1.51% respectively) as well as a lower prevalence than those in the general population of 16% [71] and 3% [72]. Although the reason for this is unclear, we believe that perhaps anxiety disorders and psychosis are less researched in these populations due to a lack of research focus or a bias in selection of study participants. Also, according to Derek Summerfield, western notions of anxiety and psychosis, and therefore the criteria and tools used to diagnose them, may not be applicable to African settings [73].

A number of the predictors of MH disorders identified in this study, such as socio-demographic factors, social and interpersonal relationships, as well as exposure to traumatic experiences and violence, are supported by evidence in the literature [74]. PTSD, the most frequently studied condition, is present in more than 50% of these groups, due to their exposure to severe traumatic experiences related to displacement in conflict-ridden African countries.

Our findings highlight specific gender patterns. The majority of asylum seekers, migrants, and refugees leaving Africa are young males. Furthermore, females are more likely to suffer from PTSD than males due to the significant trauma they have experienced due to gender-based sexual violence. According to studies on PTSD and gender, women are more likely than

men to suffer from PTSD [75, 76]. It is more likely that women will experience sexual violence during times of conflict [77, 78], which may increase their risk of PTSD, as well as childrearing pressures, safety concerns, and exploitation. Despite this, studies from countries that have been reported to commit systematic sexual violence do not adequately describe the type of trauma associated with PTSD diagnosis [79]. Furthermore, there are major gaps in the literature when it comes to disaggregating data by gender for other mental health conditions, whereas gender disaggregated data for PTSD is readily available. According to best reporting practices, future research in this field should disaggregate outcome measures by gender.

In accordance with findings from a global systematic review in 2020 [80], our review also identified several predictors of post-migration mental health disorders in host countries, including prolonged detention, long waits for refugee permits, limited healthcare access, acculturation difficulties, and low social support and integration. The West underestimates these challenges, as do African nations. The prevalence of postmigration mental health disorders is higher in countries with harsh immigration policies, such as detention, deportation, and delays in refugee determination and resettlement processes. Likewise, local population shifts against immigration host countries and increased hostility towards refugee populations negatively impact social integration and increase the risk of mental illness. Furthermore, MH disorders in asylum seekers, migrants, and refugees' groups can have long-lasting impacts on the next generation. It appears that there are gaps in the existing literature, such as the lack of longitudinal studies in these groups and assessments of the economic costs associated with MH disorders in African populations.

## MH disorders are more prevalent among children, adolescents, and their families from African asylum seekers, migrants, and refugees

The prevalence of depression, anxiety, emotional problems, and behavioral problems among refugee and asylum seeker children and adolescents was generally higher than that of native children and adolescents [81–84]. It is possible for these differences to vary depending on the type of trauma experienced in the country of origin or during migration, as well as the challenges and problems faced by the host country [85]. The development of psychiatric disorders in children and adolescents is affected by several factors [85–87]. Among them: pre-migration factors such as poverty, violence, wars, warlike conditions, education, family values, social and cultural values; peri-migration issues such as separation, abuse, and trafficking; and post-migration issues such as schooling, social support, stable settlement, parental mental health, and legal aspects of immigration. This scoping review found that unaccompanied minors suffer from more anxiety disorders, depressions, and PTSD than accompanied minors [88], which indicates that they require special care. As a result, while this review summarizes migration factors' associations with mental health disorders, it does not provide an in-depth analysis of this complex interplay.

The prevalence of PTSD and depression in the postmigration environment suggests that ongoing stressors may contribute to MH problems. In particular, Anakwenze and Rasmussen's [29] study indicates that mental illness in parents of asylum seekers, migrants, and refugees' groups can manifest in their children, who are born and raised there without having to deal with displaced conditions or volition in their original countries. Foreign environments can present several challenges, including social and cultural isolation, reorganization of family relationships, difficulty adjusting to the culture, and limited economic and social opportunities.

MH in postmigration environments may also be hampered by low social support, accumulation challenges, and other factors. In contrast to depression or post-traumatic stress disorder, displaced individuals were more likely to experience anxiety. Fazel and colleagues' study on the

mental health of displaced and refugee children resettled in high-income countries [85] reported a higher prevalence of posttraumatic stress disorder (PTSD) and depression in children. It is likely that this is due to the fact that we included studies of refugees' populations in low- and middle-income countries, where there may be a greater number of predictors and risk factors. Another explanation is based on methodology. It is likely that there is a difference in prevalence because it was not aggregated into a single figure, as in systematic reviews, but described as a range of values from individual studies that were highly heterogeneous in nature.

## Literature gaps regarding African asylum seekers, migrants, and refugees

During our review, we identified that there is a lack of comprehensive assessment of the economic burden of MH diseases among asylum seekers, migrants, and refugees, especially in African countries. In the literature, there are only a few studies that examine the economic impact of restructuring on the health sector, including mental health services, without considering the burden on those services. The absence of empirical studies on the economic dimensions of mental health in African asylum seekers, migrants, and refugees underscores the urgent need to understand the intricate relationship between mental health challenges and economic outcomes. Accordingly, future research should focus on unraveling the nuanced complexities of how mental health issues manifest and influence the economic landscape of African asylum seekers, migrants, and refugees. Additionally, a deeper understanding of these dynamics may assist us in developing supportive interventions that are tailored to these groups' specific needs.

Furthermore, in studies using certain diagnostic assessments such as the PDS, PTSD and depression were more prevalent than in studies using the ITQ or PHQ. It should be noted that most of these instruments are self-admitted; however, the PDS is detailed and includes sub-scale scores for intrusions, avoidance, and arousal symptoms, so it can report a higher prevalence rate. The same trend can be observed in instruments for other conditions as well. Overall, we found no difference between interviews conducted with interpretation assistance and those conducted in a native language. As a result of my study, it was found that although the diagnostic measures are perceived to be similar in their accuracy and performance, they differ in certain respects. Therefore, it is critical that the method and instrument used to assess refugee populations' mental health be carefully selected. Due to clinicians' reliance on self-reported symptoms when diagnosing mental illness, this difference may be caused by a number of factors, including language proficiency. It is necessary to conduct research to determine whether native interviewers conducting interpreter-assisted interviews and clinicians conducting assessments in their native language can grasp cultural and linguistic nuances that may affect diagnostic accuracy. It is important to note that we did not examine how diagnostic measures and tools may affect diagnosis of cases. In these studies, we acknowledge that western representations and understandings of psychiatric conditions have a great deal of influence on the tools developed. Consequently, either African populations' mental health disorders are overmedicalized or underdiagnosed. It is therefore necessary to conduct further research in this area.

## Strengths and limitations of the study

This study examined the prevalence, predictors, and economic burden of mental health disorders in this population and extends beyond post-traumatic stress disorder (PTSD) to include depression, anxiety, and psychosis. By specifically focusing on the African demographic, this study significantly contributes to the existing body of evidence regarding mental health disorders among asylum seekers, refugees, and migrants. This study sheds light on the substantial

presence of African asylum seekers, migrants, and refugees in economically disadvantaged African nations, underscoring the urgent need for enhanced healthcare support in these regions in contrast to many studies conducted in high-income countries. Additionally, by emphasizing the significant mental health burden experienced by asylum seekers, migrants, and refugees, as well as the burgeoning displaced population, the study underscores the necessity for both short-term and long-term mental health services following resettlement, emphasizing the importance of comprehensive mental health support to address these pressing needs.

Similarly, our study has several limitations. As seen in our review and other systematic reviews, findings can vary considerably due to differences in methodological and clinical quality among included studies. Different ethnic groups in Africa, varying ages, and the use of a variety of diagnostic tools may all contribute to this heterogeneity. Despite this high degree of heterogeneity, as is expected when investigating and analyzing prevalence across the population of global refugees, a sense of accomplishment has been achieved in knowing that current prevalence estimates can provide a starting point for further research.

Specifically, our study revealed that self-reports and proxy-reports often indicated a higher point prevalence of mental health disorders than structured clinical interviews using standard diagnostic criteria, such as DSM. A psychological or psychiatric assessment may be necessary in this case. In addition, we did not conduct subgroup analyses or exclude studies reporting data on specific subgroups, such as male-female comparisons, which could have provided valuable insights into these distinctions. To enhance the accuracy and depth of our understanding of MH disorders among asylum seekers, migrants, and refugees from African populations within the context of our study, it is important to standardize assessment methods and consider subgroup differences. It is also possible that this variation may hinder the comparability of data among countries and international organizations. To enhance data exchange and cooperation among entities such as UNHCR, a unified methodology approach should be established, incorporating clear definitions and guidelines.

## Conclusions

In summary, MH disorders are significantly more prevalent among asylum seekers, migrants, and refugees from African countries than in the general population. There are several predictors of MH disorders in this population, including sociodemographic factors and postmigration stressors, both in host African countries as well as Western host countries. The economic burden of MH disorders in this group is largely unknown. An understanding of the longitudinal effects of migration on MH disorders in this population, and the resulting economic burden, requires more research.

### Recommendations

- Prioritize mental health: Research should focus on mental health concerns among refugees and asylum seekers, emphasizing that mental well-being needs to be prioritized over physical health.

- Special attention to women and unaccompanied minors: women and unaccompanied minors, considered high-risk groups, require special attention. In addition to expediting their asylum claims, host countries should offer them employment and educational opportunities. The psychological intervention that enhances coping strategies, self-esteem, and identity rebuilding is especially beneficial for refugee groups.

- Incorporating mental health into economic programs: Interventions such as Youth Readiness Intervention (YRI) have proven effective not just in improving refugee quality but also

in terms of generating income-producing opportunities and school retention. Hence, it is important to integrate them into economic plans that target these groups of people.

• Methodological standardization and priorities for research: Comparable and validated methods and tools should be used when studying MH in these populations. Women, unaccompanied minors, and the economic aspects of the problem are important research areas. Despite evidence that MH can manifest in offspring of these human groups and be a long-term issue, our analysis found no longitudinal studies. Research in this area is also important. Without further high-quality research that examines the different components of MH needs, culturally appropriate and effective interventions, and longitudinal mental illness trajectory, which will have a significant impact on successful integration, untreated mental illnesses will adversely affect successful integration into host communities.

## Supporting information

**S1 File. Definitions of the terms used in this study, such as asylum seekers, migrants, refugees, and unaccompanied minors, as well as other technical terms, such as prevalence and economic burden.**
(DOCX)

**S2 File. A description of the search strategy and search terms.**
(DOCX)

**S1 Checklist. Preferred Reporting Items for Systematic reviews and Meta-Analyses extension for Scoping Reviews (PRISMA-ScR) checklist.**
(PDF)

## Acknowledgments

This study was a part of a MSc research project at LJUM, UK, made possible by the generous support of UNICAF Foundation. Thus, the authors sincerely acknowledge the foundation for their invaluable support.

## Author Contributions

**Conceptualization:** Wael Osman, France Ncube, Kamil Shaaban, Alaa Dafallah.

**Formal analysis:** Wael Osman.

**Methodology:** Wael Osman.

**Supervision:** France Ncube.

**Visualization:** France Ncube.

**Writing – original draft:** Wael Osman.

**Writing – review & editing:** Wael Osman, Kamil Shaaban, Alaa Dafallah.

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
