## [Decision Letter · Decision Letter 0]

26 Jan 2024

PONE-D-23-37675Prevalence, Predictors, and Economic Burden of Mental Health Disorders Among Asylum Seekers, Refugees and Migrants from African Countries: A Scoping ReviewPLOS ONE

Dear Dr. OSMAN,

Thank you for submitting your manuscript to PLOS ONE. After careful consideration, we feel that it has merit but does not fully meet PLOS ONE’s publication criteria as it currently stands. Therefore, we invite you to submit a revised version of the manuscript that addresses the points raised during the review process. Please respond to all reviewers' comments and in particular, consider re-running your search in response to Reviewer #2's comments regarding the limitations of the review search strategy, as many relevant papers may have been overlooked.

We look forward to receiving your revised manuscript.

Kind regards,

Rebecca F. Baggaley

Academic Editor

PLOS ONE

4. We note that Figure 2 in your submission contain [map/satellite] images which may be copyrighted. All PLOS content is published under the Creative Commons Attribution License (CC BY 4.0), which means that the manuscript, images, and Supporting Information files will be freely available online, and any third party is permitted to access, download, copy, distribute, and use these materials in any way, even commercially, with proper attribution. For these reasons, we cannot publish previously copyrighted maps or satellite images created using proprietary data, such as Google software (Google Maps, Street View, and Earth). For more information, see our copyright guidelines: http://journals.plos.org/plosone/s/licenses-and-copyright.

1. You may seek permission from the original copyright holder of Figure 2 to publish the content specifically under the CC BY 4.0 license. 

Reviewers' comments:

Reviewer's Responses to Questions

**Comments to the Author**

1. Is the manuscript technically sound, and do the data support the conclusions?

Reviewer #1: Partly

Reviewer #2: Partly

2. Has the statistical analysis been performed appropriately and rigorously? 

Reviewer #1: N/A

Reviewer #2: N/A

3. Have the authors made all data underlying the findings in their manuscript fully available?

Reviewer #1: Yes

Reviewer #2: Yes

4. Is the manuscript presented in an intelligible fashion and written in standard English?

Reviewer #1: Yes

Reviewer #2: Yes

5. Review Comments to the Author

Reviewer #1: To the authors:

Thank you for the opportunity to review this piece of work. This is an important area, and there is a need for research on the mental health needs of these the communities, and the larger impacts that they have. Congratulations, too, as you mention in the manuscript that this work was carried out as part of an MSc. This is a needed area of research, and the paper presents some important findings. There are some areas which could be strengthened, which I provide suggestions for below.

Abstract

Background:

I would encourage the authors to think about wording when describing migrant populations to ensure the article is not stigmatizing. For example, rather than “Inflows of migrants, asylum seekers, and refugees from African countries pose significant health and economic burdens” consider something like “Migrants, asylum seekers, and refugees from African countries may have significant health needs, with economic implications for receiving countries globally.” I would also reconsider ‘economic burden’ as a key word.

Similarly, these populations are very heterogeneous and not all migrants are more likely to experience mental health needs (and not all of these needs are ‘disorders’) so you may consider rephrasing the second sentence as: These communities may experience mental health needs due to factors such as…”.

Objective:

In your objective I might consider using a different word than ‘determine’ as you are not doing a systematic review and meta-analysis, for example ‘examine’ or ‘explore’.

Methods:

It would be helpful to have more information about the selection criteria (inclusion/exclusion) in the abstract.

Conclusion:

I would reconsider the first sentence, as the findings don’t imply mental health disorders are a major public health problem – rather, I would say something like “the findings of this scoping review suggest that many migrants from African countries experience mental health needs.” (This avoids stigmatizing the mental health needs of these communities by labeling them as problems).

Introduction

As an overall point, it would be helpful to define the terminology you are using for these communities early on. For example, how are you distinguishing asylum seekers, refugees, and migrants? Other terms I noticed in the paper included forced migrant, immigrants, non-forced migrant populations. It would be useful to have any terms used defined, and then used consistently.

Some of your citations (e.g. 2) are nearly ten years old, so I would consider updating with more current information (particularly as migrant patterns have also changed in the context of covid).

I would consider adapting some of your wording to avoid stigmatizing these communities. For example, rather than “The ability to cope with stress, work productively and fruitfully, and contribute to society is significantly impaired…” something like “Asylum seekers, migrants, and refugee groups may experience barriers to coping with stress, working productively and fruitful, and contributing to society” as not all migrants will be affected, many can contribute meaningfully, and it is the barriers placed before them (not their fault).

Methods

Spell out acronyms (like LJMU, AIR, PTSD) the first time they are used.

Regarding the selection criteria, it would be helpful to clarify whether any publications with mixed populations (both from Africa and elsewhere) were excluded, or just those where African populations could not be disaggregated. If it is the former, this should be addressed as an important limitation as there are likely to be numerous papers that do report on African populations that thus aren’t considered, which may bias the findings.

There is a section that reads “To chart the data, the first step was to review the abstracts” – I wasn’t sure if you were talking about data extraction here or screening. I do see that you have a separate section ‘data charting’ later on. Is this section perhapsmeant to be about screening? (e.g. you mean that to screen yielded papers for relevant texts an abstract screening was carried out before a full text screening?) If so, this language is probably more typical to use than “to chart the data”.

Looking at the search terms, it appears that the population terms you used were just relevant to refugees and asylum seekers. If this is the case, then the search would just have been focused on yielding papers on asylum seekers and refugees (not other migrant groups). Is this review specifically focused on these populations only? If so, this needs to be made clear throughout, including the abstract, introduction, methods (e.g. inclusion criteria) etc. In the list of search terms I also note that terms for depression, ptsd etc are not included. I would encourage the authors to consider revising the search strategy and updating the search before publication to include a broader range of mental health terms (and migrant terms if the review is intended to include individuals outside of asylum seekers and refugees). I also think there are limitations to having included search terms for Africa. I appreciate this is the population focus of the review, but the limitation in including this in the search strategy is that if, for example, a paper is specifically on refugees from South Sudan, the word Africa may not be used, but rather South Sudan. As a result, this paper may not get picked up. In my experience, thus, it is best practice to hand search by country/region. An alternative is to include all African countries and other terms (e.g. sub-saharan) in the search strategy.

For me, this guidance also applies to the use of economic terms, as it is challenging to have a comprehensive enough search strategy to identify papers that include data on economic impact, and is better to do through hand searching. In addition, my other concern is that the structure of the search strategy means that papers would have to include both mental health and economic terms to be picked up, but there is likely a wider range of literature that reports on one or the other. As a result, I think there is a risk you have missed papers. This is a challenge of trying to create one search strategy for multiple outcomes, and typically multiple search strings might be used then the results all pooled before screening.

You note that you use a modified version of the Appraisal Tool for Cross-Sectional Studies (AXIS). Is being a cross-sectional study one of the inclusion criteria? I don’t see this mentioned in the methods, so it would be helpful to clarify what study designs were included and whether only cross-sectional studies were included. If other study types were not excluded, then it raises the question of how quality was assessed for those study types. I can see in the results, for example, that qualitative research was included. This needs to be mentioned in the methods, and appropriate quality appraisal carried out.

In the ‘data charting’ section you outline what information was extracted on measurement of mental health, however you don’t mention what (if any) information was extracted on economic outcomes. If this is a key focus (which it seems to be from the aims, selection criteria, and search string) it would be good to include a description of this as well.

In the data quality assurance section you talk about selection criteria, and screening to some extent. It would be useful to have separate sections for this (see comment above about ‘screening), which presumably would precede the data extraction/charting section.

In the data analysis and synthesis section you describe generating themes. However, it seems that you were excluding qualitative studies (e.g. just including cross-sectional studies), so it raises the question of why qualitative data were not included. In addition, since your research questions are in part focused on prevalence, I feel further justification is needed for why this analysis approach was used.

Results

The results begin by describing the screening process - this content might be more suitable for the methods section (the steps used, not the numbers).

Overall, the results provide a good summary of the mental health needs identified, and key predictors, as well as highlighting the lack of data on economic impact. The tables and figures support the findings well. However, in the methods you mention that you identified themes in your analysis. I don’t see themes presented here in the results, so it may be beneficial to slightly rethink how the methods describes the process of synthesizing the information across the studies if themes were not identified and are not presented in your results.

Discussion:

The discussion highlights some important areas. Some of this could perhaps be brought out more in the results, for example differences in rates of mental illness between children and adolescents from migrant backgrounds compared to native-born populations, differences between accompanied and unaccompanied minors, and the content around families. These findings are important, and only mentioned in the discussion, so a section in the results could be beneficial.

In a few places, the discussion could be strengthened by engaging with more of the wider literature. For example, how do the rates in African migrant/refugee/asylum seeker populations compare with rates in other migrant/refugee/asylum seeker populations? It might also be useful to include some more of the literature around acculturation, and the healthy migrant hypothesis vs the migration morbidity hypothesis.

The section in the discussion around the instruments used is good. It could be further strengthened by including a discussion of the critiques of the use of wester/biomedical instruments and mental health disorders in migrant populations, as well as critiques around normative distress vs the over medicalization of these communities (see Derek Summerfield’s writing for example).

It is great you have highlighted key strengths and limitations. Typically these are written out (rather than bullet points), but this is a stylistic issue the journal can decide on. The last ‘strength’ could also be reworded to avoid stigmatizing these communities – I think it is problematic to say that these communities pose a ‘global public health threat’ – perhaps something instead like ‘these communities experience important mental health needs’. In the limitations you might also touch on the limitations of the use of these instruments in these populations.

The recommendations provide some great ideas. I would encourage the authors, though, to ensure the recommendations stem directly from what is presented in the results section. If any do not, then either remove them, tweak them, or add the relevant content that supports them to the results section.

There are some places throughout the abstract and full text where minor editing/grammar can be corrected.

Reviewer #2: Dear Authors,

Interesting research topic, but the review study needs a bit more in-depth analysis to enhance its impact. It could also benefit from reorganising the text and proofreading, as certain parts were hard to follow and lacked justifications.

Introduction

Provide a description of “forced migrants”.

Line 91- Provide more context about this “increasing the challenge of screening health conditions efficiently.” The influx of migrants is not the direct factor affecting screening efficiency.

Paragraph 2 is a bit hard to follow. The first three lines can be revised to make the point more straightforward.

Line 104-106; supporting references required.

Line 120-122: Is the scoping reviewed referred to here about general or forced migrant populations?

Lines 122-123; supporting references required.

Methodology

Why was the search date started on 1 January 2000? Provide justification.

Table 1: The search terms used in 1-4 are limiting and would have missed other papers mentioning just mentioning “immigrants” or “asylum” as a word (e.g. “people seeking asylum”).

Also, words in 19-21 would miss out on studies mentioning “finance” and “income” related terms etc, but not “economics”.

This section should clearly define the core terms “Prevalence, Predictors, and Economic Burden” in the context of this work, so it is clear precisely what you are looking into.

AXIS appraisal is for cross-sectional studies (a quantitative study design); although this was modified to 5 questions, I do not think it is appropriate to have used for the qualitative studies.

Results

Line 255 – Did all the quantitative studies use a cross-sectional design? Make this clear.

When mentioning authors in-text, you do not need to include their initials.

Paragraph 3, line 6: what comorbid conditions?

Analysis of your findings is quite limited and contrary to the extended relevance presented in the discussion. Also, there are several contextual repetitions, especially about PTSD. Further analysis by population groups (e.g. Children and adolescents) would improve the summarisation.

It is possible that the lack of relevant papers on the economic burden of MH disorders might be due to the search terms being too narrow. Most papers may not have used the word “economic.”

Discussion

Most of the information in the section “What do the review's findings mean for host countries?” is not entirely relevant to the focus of your review. The section also does not emphasise much on the relevance of your findings to host countries.

Several claims in this section do not come through clearly in your results. I recommend reanalysing your data and mapping your discussion to reflect that.

Conclusion

Ideally, the end of a journal paper ends with a conclusion statement/paragraph, not a list of recommendations. Also, the recommendations should come before the conclusion and could be placed in a text box to provide clearer emphasis.

6. PLOS authors have the option to publish the peer review history of their article (what does this mean?). If published, this will include your full peer review and any attached files.

Reviewer #1: **Yes: **Laura B. Nellums

Reviewer #2: No

---

## [Author Response · Author response to Decision Letter 0]

2 Apr 2024

Response to reviewers

Manuscript: PONE-D-23-37675

Title: Prevalence, Predictors, and Economic Burden of Mental Health Disorders Among Asylum Seekers, Refugees and Migrants from African Countries

Reviewer #1: 

To the authors: Thank you for the opportunity to review this piece of work. This is an important area, and there is a need for research on the mental health needs of these the communities, and the larger impacts that they have. Congratulations, too, as you mention in the manuscript that this work was carried out as part of an MSc. This is a needed area of research, and the paper presents some important findings. There are some areas which could be strengthened, which I provide suggestions for below.

Response:

Thank you very much, Prof. Nellums, for taking the time to review our manuscript. We really appreciate your valuable comments which indeed made our manuscript improve significantly. 

Abstract

1) Background: I would encourage the authors to think about wording when describing migrant populations to ensure the article is not stigmatizing. For example, rather than “Inflows of migrants, asylum seekers, and refugees from African countries pose significant health and economic burdens” consider something like “Migrants, asylum seekers, and refugees from African countries may have significant health needs, with economic implications for receiving countries globally.” I would also reconsider ‘economic burden’ as a key word. Similarly, these populations are very heterogeneous and not all migrants are more likely to experience mental health needs (and not all of these needs are ‘disorders’) so you may consider rephrasing the second sentence as: These communities may experience mental health needs due to factors such as…”.

Response:

Concerning these comments, we made the changes as you suggested. 

2) Objective: In your objective I might consider using a different word than ‘determine’ as you are not doing a systematic review and meta-analysis, for example ‘examine’ or ‘explore’.

Response:

Thank you for your comment. The word determine has been changed to explore.

3) Methods: It would be helpful to have more information about the selection criteria (inclusion/exclusion) in the abstract.

Response:

As a result of word count constraints, we did not mention this part at the beginning. This abstract now contains inclusion, exclusion, and value-added information. 

4) Conclusion: I would reconsider the first sentence, as the findings don’t imply mental health disorders are a major public health problem – rather, I would say something like “the findings of this scoping review suggest that many migrants from African countries experience mental health needs.” (This avoids stigmatizing the mental health needs of these communities by labeling them as problems).

Response: 

Thank you so much for your advice. To avoid stigmatizing mental health, the sentence was also rephrased. 

Introduction: 

5) As an overall point, it would be helpful to define the terminology you are using for these communities early on. For example, how are you distinguishing asylum seekers, refugees, and migrants? Other terms I noticed in the paper included forced migrant, immigrants, non-forced migrant populations. It would be useful to have any terms used defined, and then used consistently. 

Response:

There was a full table of definitions included in the MSc dissertations. As a follow-up, we have added a paragraph discussing how these concepts are defined (page 5, lines 89-102) along with a reference (IOM, reference number 3). Furthermore, we added detailed definitions for those terms as "Glossary of Migration from International Migration Law No. 34 - Glossary on Migration" in a new supplementary material file that also includes the new searching strategy. 

6) Some of your citations (e.g. 2) are nearly ten years old, so I would consider updating with more current information (particularly as migrant patterns have also changed in the context of covid).

Response:

We updated this reference, along with similar ones that had old data, with the most recent references. 

7) I would consider adapting some of your wording to avoid stigmatizing these communities. For example, rather than “The ability to cope with stress, work productively and fruitfully, and contribute to society is significantly impaired…” something like “Asylum seekers, migrants, and refugee groups may experience barriers to coping with stress, working productively and fruitful, and contributing to society” as not all migrants will be affected, many can contribute meaningfully, and it is the barriers placed before them (not their fault).

Response:

Your thoughtful feedback is greatly appreciated, Dr. Nellums. We did not intend to stigmatize these groups in any way. As an example, the above-mentioned sentence is supported by some scientific evidence. Nevertheless, we agree that the language might be interpreted as stigmatized. As a result, we changed not only this sentence, but any similar language throughout the manuscript as well. 

Methods:

8) Spell out acronyms (like LJMU, AIR, PTSD) the first time they are used.

Response:

We spelled out all acronyms the first time we used them in the revised manuscript.

9) Regarding the selection criteria, it would be helpful to clarify whether any publications with mixed populations (both from Africa and elsewhere) were excluded, or just those where African populations could not be disaggregated. If it is the former, this should be addressed as an important limitation as there are likely to be numerous papers that do report on African populations that thus aren’t considered, which may bias the findings.

Response:

Studies containing both African and non-African populations were excluded for a variety of reasons. Often, these papers included a wide range of populations as well as a large variety of prevalence measures and contained different proportions of many populations. In such studies, the primary objective is to understand migration and refuge issues in a specific location and not specific populations. We believe that this did not result in significant missing of African populations because we focused on studies that were primarily focused on Africans. Consequently, we believe we did not miss significant studies inside the African continent, nor did we miss the major refugee-affected countries outside of the continent. Inclusion criteria here should not qualify as selection bias. 

10) There is a section that reads “To chart the data, the first step was to review the abstracts” – I wasn’t sure if you were talking about data extraction here or screening. I do see that you have a separate section ‘data charting’ later on. Is this section perhaps meant to be about screening? (e.g. you mean that to screen yielded papers for relevant texts an abstract screening was carried out before a full text screening?) If so, this language is probably more typical to use than “to chart the data”.

Response:

Thank you for pointing out the error. In this sentence, we meant to screen the data. In the revised manuscript, this was corrected. 

11) Looking at the search terms, it appears that the population terms you used were just relevant to refugees and asylum seekers. If this is the case, then the search would just have been focused on yielding papers on asylum seekers and refugees (no other migrant groups). Is this review specifically focused on these populations only? If so, this needs to be made clear throughout, including the abstract, introduction, methods (e.g. inclusion criteria) etc.

Response:

In fact, migrants were searched for in our review, which is why four papers were included in our final set. Also, we apologize for overlooking this in this table, which has been updated to include detailed search terms (Please refer to the Supplementary Material file).

12) In the list of search terms, I also note that terms for depression, ptsd etc are not included. I would encourage the authors to consider revising the search strategy and updating the search before publication to include a broader range of mental health terms (and migrant terms if the review is intended to include individuals outside of asylum seekers and refugees).

Response:

As a result, the search term was updated in accordance with this point as well as the next point, which is to search by country (Please refer to the Supplementary Material file).

13) I also think there are limitations to having included search terms for Africa. I appreciate this is the population focus of the review, but the limitation in including this in the search strategy is that if, for example, a paper is specifically on refugees from South Sudan, the word Africa may not be used, but rather South Sudan. As a result, this paper may not get picked up. In my experience, thus, it is best practice to hand search by country/region. An alternative is to include all African countries and other terms (e.g. sub-Saharan) in the search strategy.

Response:

Please see our response to the point before this point (Please refer to the Supplementary Material file).

14) For me, this guidance also applies to the use of economic terms, as it is challenging to have a comprehensive enough search strategy to identify papers that include data on economic impact and is better to do through hand searching.

Response:

Once again, thank you for your comment. Developing a comprehensive search strategy for the economic terms was extremely challenging, but we revised it and developed new search terms based on an extensive literature review. 

15) In addition, my other concern is that the structure of the search strategy means that papers would have to include both mental health and economic terms to be picked up, but there is likely a wider range of literature that reports on one or the other. As a result, I think there is a risk you have missed papers. This is a challenge of trying to create one search strategy for multiple outcomes, and typically multiple search strings might be used then the results all pooled before screening.

Response:

Our goal was to find out whether mental health has an economic impact on these populations, so we designed the search strategy accordingly. The search was broadened by adding more search terms. It is likely, however, that the search results will be largely irrelevant if we do not combine the search. Based on the search terms, we identified 15 possible articles, but only one was relevant. The same issue was reported by McDaid and Park, 2023 (https://doi.org/10.1017/gmh.2023.1), who performed a rapid scoping review on the same topic and identified only 11 articles in all populations, with only one in Africa (which was the also the only identified). At first, we considered removing this part of the manuscript, but we decided to keep it because it displays the need for further research in this field.

16) You note that you use a modified version of the Appraisal Tool for Cross-Sectional Studies (AXIS). Is being a cross-sectional study one of the inclusion criteria? I don’t see this mentioned in the methods, so it would be helpful to clarify what study designs were included and whether only cross-sectional studies were included. If other study types were not excluded, then it raises the question of how quality was assessed for those study types. I can see in the results, for example, that qualitative research was included. This needs to be mentioned in the methods, and appropriate quality appraisal carried out.

Response:

I greatly appreciate you bringing this point to our attention and apologize for not clarifying it. For qualitative studies, we actually used an appraisal criterion that was consistent with the CAPS tool. Since all 10 CAPS points are included in the AXIS tool, and due to the limited number of qualitative studies, the first author interpreted mentioning AXIS as adequate. The manuscript has been updated to reflect that CAPS tool was used for critical appraisal of qualitative studies (page 10, lines 216-218). 

17) In the ‘data charting’ section you outline what information was extracted on measurement of mental health, however you don’t mention what (if any) information was extracted on economic outcomes. If this is a key focus (which it seems to be from the aims, selection criteria, and search string) it would be good to include a description of this as well.

Response:

Unfortunately, we did not find any specific papers regarding mental health's economic burden on these African groups except one. We debated whether to include economics in this paper, but we believe including it will encourage more research from those working in this field. 

18) In the data quality assurance section, you talk about selection criteria, and screening to some extent. It would be useful to have separate sections for this (see comment above about ‘screening), which presumably would precede the data extraction/charting section.

Response:

Thank you so much for suggesting this. As a result, we have created a new section titled "Screening, quality of evidence, and critical appraisal of studies". This section includes the screening process, which was previously included in the search strategy section. 

19) In the data analysis and synthesis section you describe generating themes. However, it seems that you were excluding qualitative studies (e.g. just including cross-sectional studies), so it raises the question of why qualitative data were not included. In addition, since your research questions are in part focused on prevalence, I feel further justification is needed for why this analysis approach was used.

Response:

Quantitative studies were not excluded from the analysis for the purpose of determining prevalences. In fact, there was only one study which used both analysis methods (cross-sectional design for the prevalence and qualitative approach, reference #25). A full search strategy was updated to include all possible articles. We would like to emphasize the fact that our inclusion and exclusion criteria, including the number of participants, were also considered in screening the articles. In some qualitative studies, there are very few participants, so they are excluded.

Results:

20) The results begin by describing the screening process - this content might be more suitable for the methods section (the steps used, not the numbers). Overall, the results provide a good summary of the mental health needs identified, and key predictors, as well as highlighting the lack of data on economic impact. The tables and figures support the findings well. However, in the methods you mention that you identified themes in your analysis. I don’t see themes presented here in the results, so it may be beneficial to slightly rethink how the methods describes the process of synthesizing the information across the studies if themes were not identified and are not presented in your results.

Response:

Your suggestions are insightful, thank you. The results and discussions were reviewed and updated according to your comment. 

Discussion:

21) The discussion highlights some important areas. Some of this could perhaps be brought out more in the results, for example differences in rates of mental illness between children and adolescents from migrant backgrounds compared to native-born populations, differences between accompanied and unaccompanied minors, and the content around families. These findings are important, and only mentioned in the discussion, so a section in the results could be beneficial.

Response:

Also, this part has been updated since our new search strategy identified two more papers on unaccompanied minors. In addition, there is now a separate section about unaccompanied minors in the Results section. 

22) In a few places, the discussion could be strengthened by engaging with more of the wider literature. For example, how do the rates in African migrant/refugee/asylum seeker populations compare with rates in other migrant/refugee/asylum seeker populations? It might also be useful to include some more of the literature around acculturation, and the healthy migrant hypothesis vs the migration morbidity hypothesis.

Response:

Thank you very much. We appreciate you being a reviewer. The discussion is updated according to the latest technolog

---

## [Decision Letter · Decision Letter 1]

31 May 2024

Prevalence, Predictors, and Economic Burden of Mental Health Disorders Among Asylum Seekers, Refugees and Migrants from African Countries: A Scoping Review

PONE-D-23-37675R1

Dear Dr. OSMAN,

We’re pleased to inform you that your manuscript has been judged scientifically suitable for publication and will be formally accepted for publication once it meets all outstanding technical requirements.

Kind regards,

Ietza Bojorquez, Ph.D.

Academic Editor

PLOS ONE

Reviewers' comments:

Reviewer's Responses to Questions

**Comments to the Author**

1. If the authors have adequately addressed your comments raised in a previous round of review and you feel that this manuscript is now acceptable for publication, you may indicate that here to bypass the “Comments to the Author” section, enter your conflict of interest statement in the “Confidential to Editor” section, and submit your "Accept" recommendation.

Reviewer #1: All comments have been addressed

2. Is the manuscript technically sound, and do the data support the conclusions?

Reviewer #1: Yes

3. Has the statistical analysis been performed appropriately and rigorously? 

Reviewer #1: N/A

4. Have the authors made all data underlying the findings in their manuscript fully available?

Reviewer #1: Yes

5. Is the manuscript presented in an intelligible fashion and written in standard English?

Reviewer #1: Yes

6. Review Comments to the Author

Reviewer #1: Thank you for all of the time and effort you have put into this. I can see how thoughtfully you addressed my comments. Well done!

7. PLOS authors have the option to publish the peer review history of their article (what does this mean?). If published, this will include your full peer review and any attached files.

Reviewer #1: **Yes: **Laura B Nellums

---

## [Editor Report · Acceptance letter]

13 Jun 2024

PONE-D-23-37675R1 

PLOS ONE

Dear Dr. OSMAN, 

I'm pleased to inform you that your manuscript has been deemed suitable for publication in PLOS ONE. Congratulations! Your manuscript is now being handed over to our production team.

Kind regards, 

on behalf of

Dr. Ietza Bojorquez 

Academic Editor

PLOS ONE